# Translocation of Oocytic HES1 into Surrounding Cumulus Cells in Bovine: Mechanism of Cellular Interaction during IVM?

**DOI:** 10.3390/ijms241511932

**Published:** 2023-07-25

**Authors:** Ralf Pöhland, Jens Vanselow, Fabiana Melo Sterza

**Affiliations:** 1Reproductive Biology Unit, Research Institute for Farm Animal Biology (FBN), 18196 Dummerstorf, Germany; vanselow@fbn-dummerstorf.de; 2Animal Science, State University of Mato Grosso do Sul (UEMS), Aquidauana 79200-000, MS, Brazil; fabiana.sterza@uems.br

**Keywords:** oocyte, cumulus, IVM, bovine, HES1

## Abstract

HES1 (hairy and enhancer of split-1, effector of the NOTCH pathway) plays a role in oocyte maturation and has been detected so far mainly in somatic follicular cells. In this study, we aimed to investigate whether HES1 is present in both compartments of bovine cumulus oocyte complexes (COCs) and whether in vitro maturation itself has an effect on its distribution. We investigated the abundance of HES1 mRNA and protein in bovine COCs characterized by Brilliant-Cresyl-Blue (BCB) stainability by RT-PCR and immunofluorescence before and after in vitro maturation (IVM). To study the interaction of the compartments and the possible translocation of HES1, we injected GFP-HES1 mRNA into oocytes before maturation and analyzed fluorescence recovery after photobleaching (FRAP). The results showed that HES1 mRNA was detectable in oocytes but not in cumulus cells. The number of transcripts increased with maturation, especially in BCB-positive oocytes. In contrast, the protein was mainly visible in cumulus cells both before and after maturation. After GFP-HES1-mRNA injection into oocytes, a signal could be detected not only in the oocytes but also in cumulus cells. Our result shows a nearly exclusive distribution of HES1 mRNA and protein in oocytes and cumulus cells, respectively, that might be explained by the transfer of the protein from the oocyte into cumulus cells.

## 1. Introduction

The final maturation of oocytes is a highly complex phenomenon. Not only the completion of meiosis but also cytoplasmic and metabolic processes prepare the oocyte for fertilization and the first steps of embryonic division. Oocytes are in close morphological and functional contact with the surrounding somatic compartments, primarily the cumulus cells [1]. Intercellular contacts of various kinds are a prerequisite for functional and temporal synchronization between the cells [2]. Clearly visible cellular extensions (filopodia) of the cumulus cells penetrate the zona pellucida and directly contact the oocyte, providing a tighter and enlarged surface. This is important not only for the transport of substances but also for cell surface receptors and the bidirectional flow of information [3].

A common and highly conserved signal transduction pathway that is activated via direct cell-to-cell contacts is the NOTCH signal pathway. It has already been shown in mammalian ovaries [4,5] that this pathway is important for the recruitment of primordial follicles [6,7] during folliculogenesis [8], as well as for oocyte development and the proliferation and function of granulosa cells [9,10,11,12]. The expression of the components of this signaling pathway is highly dynamic and, among others, depends on the oestrous cycle [13]. The oocyte itself obviously plays an important role (as a trigger) in the activation of the Notch signaling pathway in granulosa cells [14]. In general, it seems that the oocyte itself determines the developmental speed and synchronicity with the granulosa cells [3]. Thus, the NOTCH signaling pathway and its downstream effectors seem essentially involved in the regulation and control of folliculogenesis and oocyte development.

Among the target effectors, proteins of the HES family are already known for their repressive effect on the transcription of differentiation-relevant genes. These transcription factors are effective under the control of the NOTCH pathway [15,16]. It has been shown that, at least in the mouse, HES1, which is located in the somatic compartments of the ovary, is essential for the maturation and development of oocytes in the ovary [17].

Most of the previous investigations were carried out in murine systems. The main focus was on knockout mice, gene expression studies, and immunohistochemical protein detection in the ovaries. Investigations in other species, e.g., in the bovine system, are very rare. The same is true for in vitro matured COCs. We therefore wanted to investigate whether the downstream effector HES1 is detectable in the two compartments (oocyte and cumulus) of in vitro matured bovine COCs and whether the in vitro maturation itself has an influence on HES1 expression, distribution, and transport. In addition, the COCs were characterized by brilliant cresyl blue staining (BCB staining). This method is, in the correct sense, an activity staining for glucose-6-phosphate dehydrogenase and indicates an essential and necessary change in the energy metabolism of oocytes. These changes are important in preparation for early embryonic development and thus represent a tendency to distinguish more developmentally competent oocytes from less competent ones (with respect to early embryonic development) [18,19]. However, this staining is not exclusive in character.

## 2. Results

After BCB staining, 53.5% of the oocytes were classified as BCB+ and 46.5% as BCB−. The proportion of oocytes with excluded polar bodies after maturation was estimated. This was about 90% for both groups, with *n* = 30 each. In all further investigations, basically the same numbers of COCs were analyzed in the experimental groups.

No HES1 transcripts could be detected in the cumulus cells, neither before nor after maturation (Figure 1, left). In contrast, HES1 transcripts could be measured in oocytes (Figure 1, right), clearly increasing after maturation. This difference was more pronounced for BCB+ oocytes (0.56 to 1.16) than for BCB− oocytes (0.66 to 0.85). The differences before and after maturation and the difference between BCB+ and BCB− after maturation were significant.

In contrast to the distribution of transcripts, the HES1 protein itself, as detected by immunofluorescence, showed very strong staining in the cumulus cells (Figure 2A). Quantitative differences between the groups BCB− and BCB+ or the state before and after maturation, were not visible. The fluorescence signal from the cumulus cells was so strong that initially hardly any signal was visible in the oocytes themselves. A further increase in the amplification of the detector was necessary to see a signal in the cytoplasm of the oocyte, albeit considerably weaker (Figure 2B above). The cumulus cells were already overexposed at this amplification and did not show any structures. However, discrete cellular extensions (filopodia) are clearly visible, which reach radially through the zona pelucida. The unspecific staining (Figure 2B below) was also recorded at this high amplification.

Sixteen to eighteen hours after in vitro maturation and microinjection of HES1-GFP mRNA into the oocytes, GFP-mediated fluorescence was detected in 14 out of 49 (three experiments) COCs (28.6%). The typical green fluorescence was detectable in oocytes and cumulus cells as well (Figure 3A). No significant differences in intensity between oocytes and cumulus cells or any recognizable distribution patterns could be observed.

Six COCs with the strongest fluorescence (2 out of 3 experiments each) were used in FRAP experiments. In all cases, after bleaching, fluorescence recovery occurred both in the oocyte and in the compartment of the cumulus cells. On average, 69% ± 11% (oocytes) and 62% ± 14% (cumulus cells) of the original fluorescence intensity (after correction of unspecific bleaching) were achieved (Figure 3B). The average recovery time until the plateau phase was about the same for both compartments with 30–40 min. In some experiments a somewhat longer recovery time was observed in the compartment of the cumulus cells (e.g., Figure 3C). However, this tendency could not be statistically confirmed for all six experiments.

## 3. Discussion

Our results showed that in the bovine material examined, HES1 mRNA was detectable in oocytes but not in cumulus cells. The number of transcripts increased with maturation, especially in BCB-positive oocytes. Surprisingly, and in contrast, the protein was mainly visible in cumulus cells both before and after maturation. Possible species-specific influences must be pointed out here. In the murine system, for example, HES1 could be detected in both compartments [14].

After injection of GFP-HES1 mRNA into oocytes, the signal of GFP-tagged HES1 protein could be detected in oocytes as expected, but just additionally in cumulus cells. Finally, our results showed an almost exclusive distribution of HES1 mRNA in oocytes and protein in bovine cumulus cells.

Gene transcription is certainly a necessary prerequisite, but it does not guarantee that the corresponding functional protein is actually detectable. In oocytes during final maturation, it was found that a large number of mRNAs are produced but remain untranslated. These mRNAs form a reservoir that is used by the zygote and the early embryo until gene expression controlled by the embryo begins (in cattle after the 8-cell stage) [20,21,22]. Therefore, it is not surprising that a considerable amount of HES1 mRNA was present in the oocyte, but the associated protein could hardly be detected. The differences in the amount of transcripts before and after IVM could also be explained by this phenomenon. Especially in maturing oocytes it is therefore questionable to deduce a certain function from the sole presence of specific transcripts. In the case of HES1, it is therefore conceivable that the oocyte will retain this mRNA for later use of HES1 during early embryonic development and that only a minimum of translation will occur. This also explains the slightly higher transcript abundance in BCB+ oocytes, which are the developmentally most competent oocytes [18,19], after vs. before maturation. When considering solely and exclusively the oocytes, these explanations are logical and obvious. However, the situation in the surrounding cumulus cells is considerably more complex.

To our surprise, we could not detect significant amounts of HES1 mRNA in the cumulus cells, despite clearly detectable HES1 protein. While considering only the cumulus cells, this phenomenon can hardly be explained. Although methodological problems can never be completely excluded, they appear unlikely since the respective method worked as expected in the other compartment of the same experimental approach. One possible explanation could be that transcription and translation of HES1 in cumulus cells could have taken place earlier during follicular maturation in the ovary and only the protein is still present due to its longer half-life as compared with the mRNA. However, according to our results, mRNA transcription would have to be stopped during the follicular maturation wave, sometime before COC extraction, since even in COCs immediately after isolation, no corresponding mRNA could be detected in the cumulus cells. Since we worked with unsynchronized material from the slaughterhouse, it must be assumed that we used material from different days of the estrous cycle and thus different stages of follicle maturation. Since we never found HES1 mRNA in cumulus cells, this hypothesis seems very unlikely, especially since the protein itself was always present in very high amounts.

However, if we consider the cumulus-oocyte complex as a functional unit, the nearly exclusive presence of HES1 mRNA in the oocyte and the corresponding protein in the surrounding cumulus cells clearly suggest active interactions between both compartments. According to our more recent understanding [3], it is above all the developing oocyte that determines the timing and synchronization of follicular development. It has already been discussed by others that larger molecules, such as mRNAs or proteins, can be directly exchanged through the zona pellucida between cumulus cells and the oocyte [23]. Considering the RT-PCR and immunofluorescence data of the present study, the HES1 mRNA and the protein itself could thus be actively transported between the oocyte and cumulus cells. As a functional proof, a recombinant mRNA encoding a HES1/GFP fusion protein was injected into the oocytes of unmatured COCs, which were then subjected to IVM. In the case of active gene expression, the GFP-labeled HES1 fusion protein should be detectable as a green fluorescence signal. This was actually the case in almost one-third of the injected oocytes. In this context, it must be taken into account that the injection into the oocyte cytoplasm is difficult to control under the given conditions. On the one hand, the bovine oocyte membrane is relatively elastic, and on the other hand, the surrounding cumulus cells complicate a safe injection. It is therefore possible that these injections have not been successful in all cases. Another problem is the special situation mentioned above regarding translation into oocytes. It is not certain that transcribed RNA will be translated during the final maturation of the oocyte. Under these conditions, a successful translation of the HES1-GFP fusion protein in almost one-third of the injected oocytes can be considered successful. Most interestingly, however, we noticed that in all cases of successful expression of the HES1-GFP fusion protein in the oocyte, a positive signal was also found in the surrounding cumulus cells. This is very unlikely due to involuntary carryovers of the mRNA into injured cumulus cells during the injection itself, since the signal was visible not only on one side, i.e., near the injection area, but throughout the entire cumulus. This suggests that either the mRNA injected into the oocyte or the translated protein was translocated into the cumulus cells. To verify this, FRAP experiments were performed. Both in the bleached area of the oocyte and in the bleached area of the cumulus, fluorescence signals were recovered. The kinetics of the recovery showed no significant differences between the two compartments. Only in some cases could a tendency toward a somewhat slower recovery in the cumulus cells be observed. This result is difficult to interpret. A protein synthesis in the oocyte and a subsequent translocation into the cumulus cells would lead to a much longer recovery phase in the cumulus cells compared with the oocyte. However, this was not the case. On the other hand, the 40 min recovery time for a de-novo synthesis in the cumulus cells themselves is relatively short. The most likely explanation seems to be that the extremely high mRNA concentration after injection also resulted in a correspondingly high concentration of HES1-GFP protein in the oocyte. This led to fluorescence recovery in the oocyte by the redistribution or diffusion of already existing proteins into the bleached area. At the same time, a translocation into the cumulus cells also took place, which led to a very fast recovery in the cumulus cells. However, this explanation is only a hypothesis since we cannot exclude the possibility that the injected mRNA was translocated into the cumulus cells due to its highly abundant concentration after injection. It is suggested that recent methodological advances such as single-cell RNA sequencing in individual compartments may provide more clarity [24]. It would also be useful to try to specifically inhibit a possible transfer of molecules between cells. However, this must be reserved for future work and further methodological advances.

In conclusion, HES1, as an effector of the NOTCH signal pathway, seems to play a role in isolated and in vitro matured COCs. Its regulation seems to take place as a result of a complex interaction between the two compartments, the oocyte and cumulus, with the oocyte possibly having a trigger function. A direct exchange of macromolecules, such as HES1 protein and possibly mRNA, between the compartments through the zona pellucida can be assumed. More detailed investigations on the exact mechanisms and their significance for the final maturation of the oocyte must follow.

## 4. Materials and Methods

### 4.1. Oocyte Recovery, Selection, and In Vitro Maturation

Bovine ovaries were recovered at a local commercial slaughterhouse in PBS and transported to the laboratory within 2 h (antibiotics, 37 °C). Follicles larger than 3 mm were aspirated with a syringe (18 G needle) and collected in petri dishes (PBS, 0.3% *w*/*v* BSA, pyruvate, heparin, penicillin and streptomycin). Only COCs with a compact cumulus classified as grade 1 or 2 were used.

Brilliant-Cresyl-Blue (BCB) staining of COCs was carried out in accordance with Alm et al., (2005). BCB stained (BCB+) and unstained (BCB−) COCs were matured separately in groups (20 to 30 per well) in 4-well plates containing 420 µL of maturation medium (TCM199 with Earl salts, 5% *v*/*v* estrus cow serum, 0.5 ng/mL estradiol, 0.01 mIU/mL HCG, 200 mM L-glutamine, 0.01 mg/mL streptomycin, 10 U/mL penicillin) under mineral oil in an incubator (HeracellTM, ThermoScientific, Bonn, Germany) with 5% CO_2_ at 38.5 °C for 24 h. A corresponding group of COCs was not mature and served as a control.

### 4.2. DNA and RNA Preparation, cDNA Synthesis, and Real-Time Quantitative PCR

RNA from oocytes and cumulus cells was isolated separately with a Nucleus Spin RNA II kit (Macherey-Nagel, Düren, Germany) following the manufacturer’s instructions. Oocytes were separated from cumulus cells by frequent pipetting (mechanical separation) of COCs. Oocytes were collected and washed (3×) in PBS. 30 Oocytes per group (3 replicates) were transferred with as little as possible (<5 μL) of PBS to the lysis buffer. The corresponding cumulus cells were pooled, washed (3×) in PBS, and re-suspended in lysis buffer. RNA concentration was measured with a NanoDrop1000 spectrophotometer (ThermoScientific, Bonn, Germany). The cDNA synthesis was performed with M-ML reverse transcriptase and ribonucleaseRNase inhibitors (M3683 and N2515, Promega Mannheim, Gemany) using oligo- (dT) primers (2 ng/μL, #10814270001) mixed with random hexamer primers (4 ng/μL, #11034731001, Roche, Mannheim, Germany) according to the manufacturer’s instructions. Subsequently, the cDNA was cleaned with the high-purity PCR purification kit (#11732676001, Roche, Mannheim, Germany) and eluted in 115 μL of elution buffer.

The real-time PCR was carried out with SensiFast™ SYBR No-ROX (#BIO-98020, Bioline, Luckenwalde, Germany) and gene-specific primers (Table 1). The abundance of each transcript was quantitated in triplicate in a LightCycler^®^ 480 instrument (Roche, Mannheim, Germany) with the following cycle conditions: preincubation at 95 °C for 5 min, 40 denaturation cycles at 95 °C for 20 s, annealing at 60 °C for 15 s, extension at 72 °C for 15 s and acquisition of single-point fluorescence for 10 s. The analysis of the melting points of all samples was completed to ensure the amplification of the correct products. The length of the PCR products was checked after each run by agarose gel electrophoresis (3%, stained with ethidium bromide). Initially, all amplicons were sequenced to verify authenticity. The cloned PCR products of the studied transcripts were used as external standards. Fresh dilutions of these were used to obtain five different concentrations of standards (5 × 10^−12^–5 × 10^−16^ g DNA/reaction) that were co-amplified. Levels of transcript abundance were generally normalized to the transcripts of RPS18.

### 4.3. Immunofluorescence Staining and Confocal Microscopy

Oocytes and cumulus cells from immature and 24-hour-matured COCs were partially but not completely separated by frequent pipetting under microscopic control. The complexes (10 per group, 3 replicates) were then washed (3×) in PBS and fixed in 3% *v*/*v* paraformaldehyde (+2% *w*/*v* Sucrose in PBS, 4 °C, 4 h).

In principle, immunofluorescence staining and confocal laser scanning microscopy were performed as already described in Pöhland et al. [25]. After washing (wash buffer: TBS + 0.5% *v*/*v* + 0.05% *v*/*v* Triton X100) for 10 min (room temperature), the complexes were permeabilized (permeabilizing buffer: 20 mM HEPES, 300 mM sucrose, 50 mM NaCl, 3 mM MgCl2, 0.5% *v*/*v* Triton X100, pH 7.4) for 10 min at, 0 °C), washed again in wash buffer for 5 min, and incubated with Roti^®^ Block (Carl Roth, Karlsruhe, Germany; 1:50 in H_2_O, 2 h, room temperature) to block non-specific binding. The complexes were then incubated with the first antibody (monoclonal mouse anti HES1, C-terminal part, IgG1, mab0077, Covalab, Cambridge, UK) diluted in TBS (0.05% *v*/*v* Tween 20, 2% *v*/*v* Roti^®^ Block) at 4 °C overnight, washed in wash buffer (4 × 10 min, room temperature), and incubated with the second antibody (goat anti mouse Alexa^®^ 488, A-32723, Invitrogene/Thermo Fisher Scientific, Dreieich, Germany) for 5 h (dark, room temperature). After washing (wash buffer, 4× 10 min, room temperature), a second fixation (2% *v*/*v* paraformaldehyde in PBS, 4 °C overnight) was carried out, the complexes were mounted with glycerin gelatin on cover slides, cooled, and stored at 4 °C until confocal microscopy.

Staining was then evaluated in a confocal laser scanning microscope (LSM 5 Pascal) attached to an Axiovert 200 M inverted microscope (Carl Zeiss, Jena, Germany). For the measurement of the green fluorescence, single-track procedures were used [488 nm of an argon laser (458, 488, 514 nm; 5 mW), 500–540 nm filter]. Stacks of images (40 images, Δz = 4 μm, 1024 × 1024 pixels) were recorded using a 40× lens (oil immersion) and the software provided by Carl Zeiss (Zen, vers. 2.1). The distribution of the fluorescence staining was studied in the different compartments (oocytes and cumulus cells) individually by sieving the stacks of images. As a representative approach, a level was selected on the complexes, which revealed oocytes with the largest diameter (medium).

### 4.4. Expession of a HES1/GFP Fusion Protein and Fluorescence Recovery after Photo Bleaching (FRAP)

The lyophilized commercial plasmid RG211709 (Origene, Herford, Germany), including a CMV promoter driven human HES1/GFP fusion gene, was dissolved in 10 µM Tris buffer (0.5 µg/µL) and transformed into competent XL1 blue cells. After picking a single colony, the bacteria were eventually grown in 100 mL LB-medium with Ampicillin and the plasmid DNA was recovered and purified using the EndoFree Plasmid Maxi Kit (Qiagene, Hilden, Germany) according to the manufacturer’s specifications. RNAse was removed with proteinase K digestion (200 µg/mL, #AM2546, Ambion/Thermo Fisher Scientific, Dreieich, Germany) at 50 °C for 90 min and subsequent phenol/chloroform extraction. After linearization with StuI a mMESSAGE mMACHINE^®^ T7 Ultra Kit (AM1345, Thermo Fisher Scientific, Dreieich, Germany) was used for in vitro transcription according to the manufacturers guidelines. After purification with phenol/chloroform, the RNA was precipitated with isopropanol, dissolved in nuclease-free water, and quantificated in a NanoDrop1000 spectrophotometer (ThermoScientific, Bonn, Germany). For injection, the RNA was diluted to 1 µg/µL RNA and stored at −70 °C until injection.

After COC recovery and washing (see above), a large portion but not all of the cumulus cells were removed by repeated rapid but also careful pipetting in the wash medium. Oocytes with a suitable number of cumulus cells (3–10 layers, no visible damage) were selected and transferred into droplets (8 µL wash medium under oil, 10 COCs per droplet, 4 droplets per tray) into an injection tray (lid of a Petri dish, 3.5 cm, Falcon, Thermo Fisher Scientific, Dreieich, Germany). A micromanipulation device (TansferMan/TransferMan MK) consisting of a holding pipette (VacuTip I) and an injection capillary (FemtoTip) and an injection device Femtojet 4i (all Eppendorf, Hamburg, Germany) on an inverted microscope Diaphot 200 (Nikon, Düsseldorf, Germany) was used for microinjection of the RNA construct. The FemtoTip capillary was filled with 5µL RNA solution (see above) using a microloader (Eppendorf, Hamburg, Germany). After insertion into the manipulator, the capillaries were checked for possible bubbles or blockages. Each individual COC was first carefully docked to the holding pipette in such a way that the cumulus was not damaged or even sucked off. The orientation was such that an area with as few cumulus cells as possible was positioned exactly opposite the holding pipette. At this point, the injection capillary was inserted centrally into the cytoplasm of the oocyte. The injection was performed with FemtoJet for an injection time of ti = 0.15 s, an injection pressure of pi = 95 hPa, and a holding pressure of pc = 20 hPa. Immediately afterwards, the injection cannula was pulled out, and the complex was released by the holding pipette. The injection of 4× 10 oocytes per Petri dish did not last more than 40 min. Immediately afterwards, all samples were transferred into maturation medium, and normal IVM was started (see above).

After an in vitro maturation for 16–18 h (see above), the COCs were individually transferred into droplets (8 µL, maturation medium under oil) in a Petri dish (lid, 6 cm, Nunc, Thermo Fisher Scientific, Dreieich, Germany). In the confocal laser scanning microscope (LSM800 + Observer.Z1, Carl Zeiss, Jena, Germany) equipped with a microscope incubator (XLMulti S1, Pecon, Ulm, Germany), the COCs were identified (under incubation conditions: 5% CO_2_, 38.5 °C saturated humidity), which showed a clear fluorescence signal (488 nm of a diode laser 510–530 nm filter).

For the FRAP investigations, two COCs with the strongest fluorescence signal were selected in three independent test runs (a total of 6 COCs). The FRAP experiments were performed in a confocal laser scanning microscope LSM800 (see above, 40× oil lens) using the software tools of the instrument manufacturer in a microscope incubator (see above). Three regions of interest (ROI) were defined at the central levels of the COCs. One ROI was placed close to the membrane above the cytoplasm of the oocyte (ROI 2), and another ROI (ROI 1) was located above the cumulus cells near the zona pelucida. A third ROI (ROI 3) to determine the unspecific fading of fluorescence during the experiment was also located near the zona pellucida above fluorescent cumulus cells (see Figure 3C left). A time series of 20 images with a resolution of 1024 × 1024 pixels was automatically recorded at intervals of 10 min. The fluorescence was bleached only in ROI 1 and ROI 2 after the acquisition of the first image with 700 iterations at full laser power. The evaluation was performed by calculating the relative integral fluorescence activity per ROI in relation to the respective integral fluorescence intensity in ROI 3 (fading control) at the respective time by the software of the device manufacturer (see Figure 3C right). For the calculation of the mean curves from all six individual experiments, the measured values were exported and calculated in R (3.6.3, open-source) as mean values and STD.

### 4.5. Statistic

The R (3.6.3, open-source) software package was used to calculate the average fluorescence intensities and analyze the data. The normal distribution of the data was evaluated by the Shapiro-Wilk test. For parametric data, an ANOVA was performed (if possible), and when a significant difference was found, the t-Test or Tukey test with a 5% probability was performed. For non-parametric data, the Kruskal-Wallis statistical test was performed with a 5% probability.

## Figures and Tables

**Figure 1 ijms-24-11932-f001:**
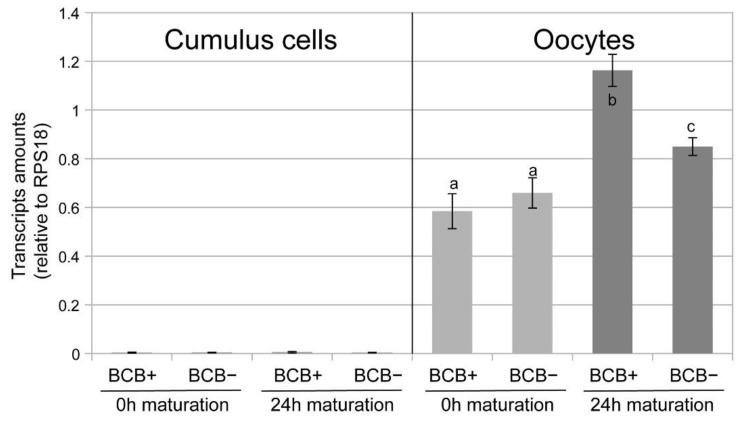
HES1 transcripts relative to the house keeping gene RPS18 measured with RT-PCR in samples of cumulus cells recovered from 30 COCs (left) and 30 denuded oocytes (right) (3 replicates) depending on BCB staining. Different letters indicate significant differences (*p* < 0.05).

**Figure 2 ijms-24-11932-f002:**
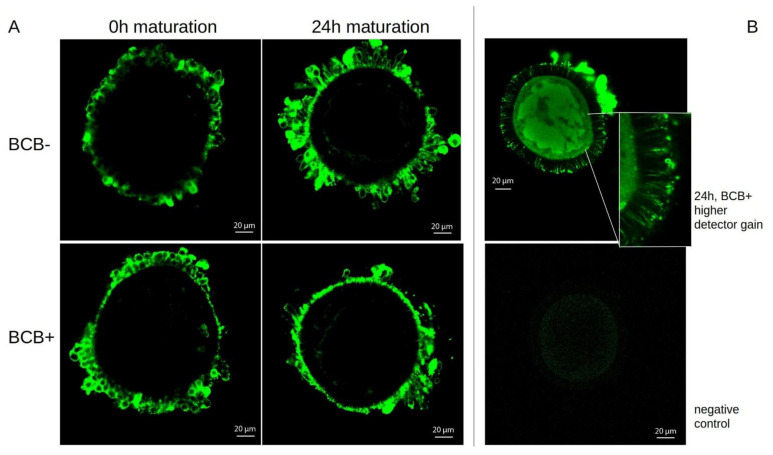
Examples of confocal microscopic images of HES1 protein detected in COCs by immunofluorescence before and after maturation depending on BCB status (**A**). With increased amplification specific fluorescence signals became visible in the oocyte as well as in cellular extensions through the zona pelucida (Example BCB+ after maturation, (**B**), top, unspecific control (**B**), bottom).

**Figure 3 ijms-24-11932-f003:**
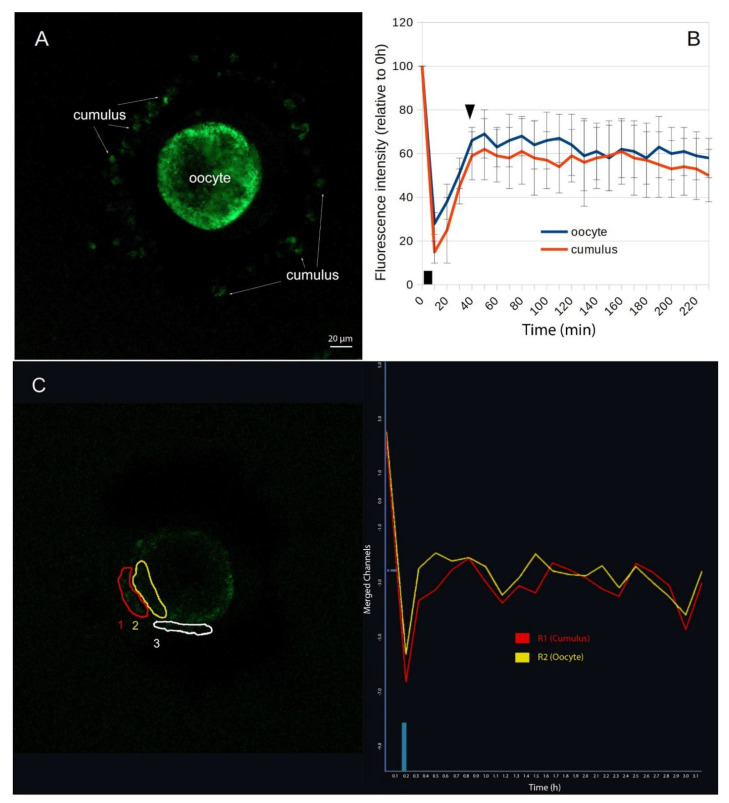
Confocal laser scanning microscopy image of a COC matured in vitro 18 h after microinjection of HES1-GFP-mRNA into the oocyte ((**A**), example). In green, the translated GFP-labelled HES1 fusion proteins can be seen in the cytoplasm of the oocyte and the surrounding cumulus cells. Fluorescence recovery after photobleaching (FRAP): The average fluorescence intensity (from 6 independent measurements/COCs) relative to the respective control measurement window is shown in an area of the ooplasm and an area with cumulus cells as a function of time (**B**). The black square symbolizes the time period of the photo bleaching, the black triangle the time of the recovery. Screenshot (ZEN 2.1, instrument software, Carl Zeiss, Jena, Germany) of an exemplary single FRAP experiment (**C**): On the left the image of an exemplary COC with the three measurement windows (1 and 2 measurement windows, 3 control windows) is shown, on the right the real-time measurement curves with the time stamp of the bleaching.

**Table 1 ijms-24-11932-t001:** Primer sequences used for rtPCR.

Name	Sequence	Bp
HES1F	TCTACACCAGCAACAGCGGGA	100
HES2R	TTCCGCCACGGTCTCCACAT	100
RPS18 forward	GAGGTGGAACGTGTGATCACCATT	
RPS18 reverse	TGTATTTCCCGTCCTTCACGTCCT	

## Data Availability

All primary research data are stored centrally at the FBN in accordance with the FBN Research Data Policy (https://www.fbn-dummerstorf.de/fileadmin/media/PDF/FBNResearchDataPolicy.pdf, accessed on 19 June 2023), the FBN open access policy (https://www.fbn-dummerstorf.de/fileadmin/media/PDF/FBNOpenAccessPolicy-en.pdf, accessed on 19 June 2023) and in compliance with the rules of the German Research Foundation (DFG) and can be made available by the corresponding (poehland@fbn-dummerstorf.de) author upon request.

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
