# Peer review of "Translocation of Oocytic HES1 into Surrounding Cumulus Cells in Bovine: Mechanism of Cellular Interaction during IVM?"

_ijms, 2023, doi:10.3390/ijms241511932_

Round 1
Reviewer 1 Report
In general, the article is well written and easy to follow. The study has shown that Hes1 transcript is mainly expressed in bovine oocytes while the protein was visible only in the cumulus cells. These are interesting results. Nevertheless, I think the authors should rewrite the aim of the study since there is no information regarding oocyte quality (BCB staining is an indicative marker, but it is not definitive – There quite a few BCB- oocytes that reach MII and form blastocysts anyway), and many assumptions are made based on descriptive parameters. There are no mechanistic results (or hypothesis in the discussion) explaining the potential role of Hes1 on bovine oocyte maturation.
Aim of the study: to investigate whether there is an influence of Hes1 on in vitro maturation (IVM) and oocyte quality – There is no information regarding oocyte quality. BCB positive is not enough since maturation was around 90% in both treatments.
Line 215: 0.014 mIE/ml HCG. Did you mean mIU/ml?
Discussion: I suggest the authors to start the discussion with a small paragraph highlighting the main finding of the study, and then begin the discussion from that point. Also, the authors should emphasize that they are showing data in bovine and discuss potential differences in the role of Hes1 between species. For instance, in mice, Hes1 protein has been detected in both oocytes and cumulus cells.
Reviewer 2 Report
The findings and hypotheses presented in this paper, "a nearly exclusive distribution of HES1 mRNA and protein in oocytes and cumulus cells, respectively, that might be explained by the transfer of the protein from the oocyte into cumulus cells," are very interesting. Unfortunately, however, there are no experiments and results that indicate that there is a transfer of the protein between oocytes and cumulus cells. Validation by such as drug treatments that inhibit the movement of proteins may also be helpful.
Round 2
Reviewer 2 Report
I was fully satisfied with the responses from the authors.